# Two-layer neural network on infinite-dimensional data: global optimization guarantee in the mean-field regime

**Naoki Nishikawa**
University of Tokyo
nishikawa-naoki259@g.ecc.u-tokyo.ac.jp

**Taiji Suzuki**
University of Tokyo
RIKEN AIP
taiji@mist.i.u-tokyo.ac.jp

**Atsushi Nitanda**
Kyushu Institute of Technology
RIKEN AIP
nitanda@ai.kyutech.ac.jp

**Denny Wu**
University of Toronto
Vector Institute
dennywu@cs.toronto.edu

## Abstract

Analysis of neural network optimization in the mean-field regime is important as the setting allows for *feature learning*. Existing theory has been developed mainly for neural networks in finite dimensions, i.e., each neuron has a finite-dimensional parameter. However, the setting of infinite-dimensional input naturally arises in machine learning problems such as nonparametric functional data analysis and graph classification. In this paper, we develop a new mean-field analysis of two-layer neural network in an *infinite-dimensional* parameter space. We first give a generalization error bound, which shows that the regularized empirical risk minimizer properly generalizes when the data size is sufficiently large, despite the neurons being infinite-dimensional. Next, we present two gradient-based optimization algorithms for infinite-dimensional mean-field networks, by extending the recently developed particle optimization framework to the infinite-dimensional setting. We show that the proposed algorithms converge to the (regularized) global optimal solution, and moreover, their rates of convergence are of polynomial order in the online setting and exponential order in the finite sample setting, respectively. To our knowledge this is the first quantitative global optimization guarantee of neural network on infinite-dimensional input and in the presence of feature learning.

## 1 Introduction

A variety of machine learning problems need to handle input from infinite-dimensional spaces. For instance, Ling and Vieu (2018); Ferraty et al. (2007) studied non-parametric function regression problems where the input is a function in an infinite-dimensional functional space, and Kriege et al. (2020) studied graph classification problems using kernel method, which can be cast as a learning problem with inputs from an infinite-dimensional reproducing kernel Hilbert space.

Among various models that deal with those infinite-dimensional input problems, neural networks are particularly interesting due to their ability to learn features and model nonlinear data. For example, Rossi and Conan-Guez (2005) proposed the Functional Multi-Layer Perceptron, where the input of functional data is converted to vector form by basis expansion and then fed into a neural network; the universal approximation and statistical consistency of the proposed model were also analyzed. Recently, Yao et al. (2021) proposed the Adaptive Functional Neural Network, which replaces the fixed basis functions in Rossi and Conan-Guez (2005) with adaptive bases, in order to leverage the power of representation (feature) learning and the flexibility of deep learning. While neural

36th Conference on Neural Information Processing Systems (NeurIPS 2022).

network based models with infinite-dimensional input have been applied in many applications, their optimization using gradient-based methods is not well-understood. In particular, no optimization guarantee has been shown for any algorithm in the presence of *feature learning* — such theoretical result can be challenging to establish even for finite-dimensional data.

It is experimentally observed that the gradient-based optimization methods can yield sufficiently small training error in training neural network models, despite the non-convexity of the landscape. Theoretical explanations of this observation often rely on *overparameterization*, that is, to consider sufficiently wide neural network compared with the data size, and the model parameterization can be divided into the *mean-field* regime (Nitanda and Suzuki, 2017; Mei et al., 2018; Chizat and Bach, 2018) and the *neural tangent kernel* regime (Jacot et al., 2018; Du et al., 2018). In this work we focus on the mean-field regime, as it captures the presence of feature learning, which is one of the main advantages of neural network (Chizat et al., 2019; Yang and Hu, 2021).

In the mean-field analysis, the dynamics of gradient descent is described by Wasserstein gradient flow in the space of probability distributions on the parameters. Mei et al. (2018) showed that the *mean-field Langevin dynamics* converges to the global optimal solution, and Hu et al. (2019) proved linear convergence with respect to the objective function with sufficiently strong KL divergence regularization. However, those studies mainly analyzed the algorithm in continuous time, and do not establish quantitative convergence rate for *discrete time and finite width* settings.

To overcome this limitation, Nitanda et al. (2021) proposed the Particle Dual Averaging (PDA) method that globally optimizes the KL-regularized objective with a *polynomial order* computational complexity, in a completely discrete time and finite width setting. Furthermore, in the case of finite-sum objective, Oko et al. (2022) proposed the Particle Stochastic Dual Coordinate Ascent (P-SDCA) method which further improves the computational complexity to an *exponential order* with respect to its outer loop iteration. Both PDA and P-SDCA employ a double-loop structure, and make use of Monte Carlo sampling from an intermediate target distribution in the inner loop. In particular, Nitanda et al. (2021) applied the convergence rate of the overdamped Langevin algorithm in Vempala and Wibisono (2019) for their optimization analysis, whereas Oko et al. (2022) also considered the Metropolis-adjusted Langevin algorithm (MALA) (Ma et al., 2019) for the inner loop sampling. However, if we naively apply these algorithms to infinite-dimensional input, then the optimization guarantee becomes meaningless due to the dimension dependence. Therefore, we need to construct a new methodology and theory for the infinite-dimensional problem.

**Our contributions.** In this work, we extend the mean-field analysis for neural network training to infinite-dimensional parameter space, which covers two-layer neural network with infinite-dimensional inputs. First, we establish a generalization error bound which entails that our model generalizes properly when the number of training data is sufficiently large. Next, we propose two gradient-based algorithms that globally optimize the KL-regularized objective, corresponding to the infinite-dimensional extension of PDA and P-SDCA, respectively.

As mentioned above, PDA and P-SDCA require Monte Carlo sampling, and in our setting sampling is performed in an infinite-dimensional Hilbert space. We therefore adapt an *infinite-dimensional gradient Langevin dynamics* (Debussche, 2011; Bréhier, 2014), the weak convergence of which has been established in Muzellec et al. (2022). Our contributions can be summarized as follows:

- We introduce two-layer neural network in the mean-field regime whose input is infinite-dimensional, and establish a generalization error bound of the regularized empirical risk minimizer.

- We propose two optimization algorithms for our infinite-dimensional neural network by incorporating the infinite-dimensional Langevin dynamics in the inner loop of PDA and P-SDCA.

- We prove that our infinite-dimensional extension of PDA achieves polynomial order convergence, and that of P-SDCA achieves exponential (outer loop) convergence even in an infinite-dimensional setting[1]. To the best of our knowledge, this is the first quantitative global optimization guarantee of neural network in the infinite-dimensional mean-field regime with the presence of feature learning.

**Other related works.** Ferré and Villa (2006) proposed the SIR-NNr, which incorporates the dimension reduction method termed "sliced inverse regression" into a neural network. In addition,

---

[1]We however note that there is an exponential dependency on the regularization parameter, similar to prior works on Langevin-based algorithms.

Rossi et al. (2005) introduced a radial-basis function network as a nonlinear model whose input is a function. As previously discussed, although statistical properties (e.g., consistency) of these models have been studied, convergence guarantee of gradient-based optimization has not been established.

## 2 Two-layer neural network with infinite-dimensional input

In this section we introduce the mean-field two-layer neural network with infinite-dimensional input. We first formulate mathematical notations and prepare assumptions required for our theoretical analyses, and then present a generalization error of the proposed model.

**Neural network model.** We consider a setting where the input is included in a (possibly infinite-dimensional) separable Hilbert space $\mathcal{H}$. Since $\mathcal{H}$ is separable, there is a complete orthonomal system $(e_j)_{j=0}^{\infty}$ for which it holds that $\mathcal{H} = \left\{ \sum_{j=0}^{\infty} a_j e_j \mid \sum_{j=0}^{\infty} a_j^2 < \infty \right\}$, and $\mathcal{H}$ is equipped with the inner product given by $\langle f, g \rangle_{\mathcal{H}} := \sum_{j=0}^{\infty} a_j b_j$, where $f = \sum_{j=0}^{\infty} a_j e_j, g = \sum_{j=0}^{\infty} b_j e_j \in \mathcal{H}$. Let $\mathcal{H}_0$ be a subset of $\mathcal{H}$ defined by $\mathcal{H}_0 := \{ 0 \cdot e_0 + \sum_{j=1}^{\infty} a_j e_j \mid \sum_{j=1}^{\infty} a_j^2 < \infty \}$. We consider a neural network model that takes an infinite-dimensional input $x \in \mathcal{X} \subseteq \mathcal{H}_0$ and gives an output $y \in \mathcal{Y} \subseteq \mathbb{R}$, where $\mathcal{X}$ and $\mathcal{Y}$ are domains of input and output, respectively. Each neuron in the network has parameter $\theta := a e_0 + w \in \mathcal{H}$ for $a \in \mathbb{R}$ and $w \in \mathcal{H}_0$, and is represented by

$$h_\theta(x) := \sigma_1(a) \sigma_2(\langle w, x \rangle_{\mathcal{H}}),$$

where $\sigma_1, \sigma_2 : \mathbb{R} \to \mathbb{R}$ are activation functions, which we assume to be bounded and smooth (such as tanh) [2]. We also denote $h_\theta(x)$ by $h(\theta, x)$. Let the number of neurons be $M$ and denote the set of parameters by $\Theta = (\theta_r)_{r=1}^{M}$. Then, in the mean-field regime, the two layer neural network model (with infinite-dimensional input) can be expressed as the average over the $M$ neurons:

$$h_\Theta(x) := \frac{1}{M} \sum_{r=1}^{M} h_{\theta_r}(x). \tag{1}$$

This can be seen as a finite sum approximation of the *integral form* $\mathbb{E}_{\theta \sim \pi}[h_\theta(x)]$ for a distribution $\pi$ on the parameter space. Indeed, if $(\theta_r)_{r=1}^{M}$ are i.i.d. realizations from $\pi$, then $h_\Theta(x)$ would converge to the integral form as $M$ increases. This viewpoint motivates us to design optimization algorithms for the distribution of parameters $\pi$ and utilize the convexity of the objective in the space of measures.

**Notations.** To introduce our method, we need to define a Gaussian process taking its value in $\mathcal{H}$. For that purpose, we define a subspace $\mathcal{H}_{K^\gamma} \subset \mathcal{H}$ for $\gamma \geq 0$ as follows. First, we define a linear operator $T_K : \mathcal{H} \to \mathcal{H}$ as $T_K f = \sum_{j=0}^{\infty} \mu_j a_j e_j$ for $f = \sum_{j=0}^{\infty} a_j e_j$, where $(\mu_j)_{j=0}^{\infty}$ corresponds to the spectrum of $T_K$ satisfying $\mu_j > 0 \ (\forall j)$ and is sorted in decreasing order ($\mu_1 \geq \mu_2 \geq \cdots$). Note that the orthonormal system $(e_j)_{j=0}^{\infty}$ is taken so that it coincides with the eigen-functions of $T_K$. Then, we define $\mathcal{H}_{K^\gamma} := T_K^{\frac{\gamma}{2}} \mathcal{H} = \{ T_K^{\frac{\gamma}{2}} h \mid h \in \mathcal{H} \}$ and we can naturally equip $\mathcal{H}_{K^\gamma}$ with a norm $\|f\|_{\mathcal{H}_{K^\gamma}} := \sum_{j=0}^{\infty} \mu_j^{-\gamma} a_j^2$ for $f = \sum_{j=0}^{\infty} a_j e_j \in \mathcal{H}_{K^\gamma}$. For concise notation we simply write $\mathcal{H}_K := \mathcal{H}_{K^1}$. Next, define an operator $A = \lambda_1 T_K^{-1}$ for a positive regularization parameter $\lambda_1 > 0$, i.e., $Af = \lambda_1 \sum_{j=0}^{\infty} \mu_j^{-1} a_j e_j$ for $f = \sum_{j=0}^{\infty} a_j e_j \in \mathcal{H}_K$, and let $\nu$ be a Gaussian measure in $\mathcal{H}$ whose mean is 0 and its covariance is $A^{-1}$ (i.e., $\langle z, x \rangle_{\mathcal{H}} \ (z \sim \nu)$ obeys the Gaussian distribution with mean 0 and covariance $\langle x, A^{-1} x \rangle_{\mathcal{H}}$ for any $x \in \mathcal{H}$)[3]. Finally, we define the Kullback-Leibler (KL) divergence from a probability measure $\nu_2$ to a probability measure $\nu_1$ that is absolutely continuous with respect to $\nu_2$ as $\mathrm{KL}(\nu_1 \| \nu_2) := \mathbb{E}_{\nu_1} \left[ \log \frac{\mathrm{d}\nu_1}{\mathrm{d}\nu_2} \right] = \mathbb{E}_{\nu_2} \left[ \frac{\mathrm{d}\nu_1}{\mathrm{d}\nu_2} \log \frac{\mathrm{d}\nu_1}{\mathrm{d}\nu_2} \right]$.

**Objective.** Now we define the objective of our optimization method. Suppose that the pairs of input and output are independently identically distributed from $\mathcal{D}$, and let $\ell(\cdot, \cdot) : \mathcal{Y} \times \mathcal{Y} \to \mathbb{R}$ be a loss function. In the mean-field regime, we optimize the following regularized risk minimization problem with respect to the distribution of parameters where the regularization term is given by the KL-divergence from the Gaussian measure $\nu$:

$$\min_{\pi \in \mathcal{P}_2} \quad \mathcal{L}(\pi) := \mathbb{E}_{(X,Y) \sim \mathcal{D}} \left[ \ell \left( \mathbb{E}_{\theta \sim \pi}[h_\theta(X)], Y \right) \right] + \lambda_2 \mathrm{KL}(\pi \| \nu), \tag{2}$$

---

[2]We focus on this model parameterization for better interpretability, although our infinite-dimensional mean-field analysis covers more general models.

[3]For precise definition of Gaussian measure in infinite-dimensional Hilbert space see Da Prato and Zabczyk (1996).

where $\lambda_2 > 0$ and $\mathcal{P}_2$ is the entire set of probability measures that satisfy $\mathbb{E}_{\theta \sim \pi}[\|\theta\|_{\mathcal{H}}^2] < \infty$ and are absolutely continuous with respect to the Gaussian measure $\nu$. We may consider the empirical measure of $n$ training data points $(x_i, y_i)_{i=1}^n$ as $\mathcal{D}$, for which the first term of the objective function is written as $\frac{1}{n} \sum_{i=1}^n \ell\left(\mathbb{E}_{\theta \sim \pi}[h_\theta(x_i)], y_i\right)$. In this setting, the problem can be regarded as a regularized empirical risk minimization problem. As for the second term, in a finite-dimensional setting where $A = \lambda_1 I$, we can easily see that the KL divergence is decomposed into the sum of $\ell^2$-regularization $\lambda_1 \mathbb{E}_{\theta \sim \pi}[\|\theta\|^2]$ and negative entropy $\mathbb{E}_{\theta \sim \pi}[\log d\pi(\theta)/d\theta]$. Here the negative entropy naturally arises from the *mean-field Langevin dynamics* (Hu et al., 2019; Nitanda et al., 2022; Chizat, 2022). Indeed, it is known that, if we optimize a loss function $L(q)$ with respect to a probability distribution $q \in \mathcal{P}_2$ by the mean-field Langevin dynamics with temperature parameter $\lambda_2$, then its stationary distribution is given by $q^* = \arg\min_{q:\text{density}} L(q) + \lambda_2 \mathbb{E}_q[\log(q)]$ which satisfies $q^* \propto \exp(-\frac{1}{\lambda_2} \frac{\delta L(q^*)}{\delta q})$, where $\frac{\delta L(q)}{\delta q} : \mathcal{H} \to \mathbb{R}$ is the derivative with respect to the distribution $q$ such that $\int \frac{\delta L(q)}{\delta q}(\theta) d(p - q)(\theta) = \lim_{\epsilon \to 0} \frac{1}{\epsilon}[L(\epsilon p + (1 - \epsilon)q) - L(q)]$ for any distribution $p \in \mathcal{P}_2$.

One important example that we can apply our model to is the *non-parametric functional regression problem* (Ling and Vieu, 2018; Ferraty et al., 2007). Suppose that the input $x$ is a function on $\mathbb{R}$ (such as time evolution of temperature) included in some reproducing kernel Hilbert space $\mathcal{H}_K$. One typical definition of $\mathcal{H}_K$ is given by $\mathcal{H}_K = T_K^{1/2} \mathcal{H}$ where $\mathcal{H} = L^2(\mu)$ for a probability measure $\mu$ and $T_K$ is an integral operator given by $T_K f(t) := \int K(t, s) f(s) d\mu(s)$ for a positive definite kernel $K$ (Caponnetto and De Vito, 2007; Steinwart and Christmann, 2008). The goal is to estimate some non-linear functional from the input $x \in \mathcal{H}_K$ to output $y \in \mathbb{R}$ (using neural network (1)).

**Generalization bound.** We now give a generalization error bound for our two-layer neural network in the empirical risk minimization setting, which confirms that the model properly generalizes when the data size is sufficiently large. Specially, we show the generalization bound for two problem settings: binary classification and regression. The following theorem stats the generalization bound for binary classification problems. We defer the regression case to Appendix A due to space limitation.

**Theorem 1.** *Let $\mathcal{D}$ be a distribution of $\mathcal{X} \times \mathcal{Y}$, $\ell$ be the smoothed hinge loss, and denote the 0-1 loss by $\ell_{01}(z, y)$, i.e., $\ell_{01}(z, y) := \mathbb{1}[zy < 0]$. Let $\pi_*$ be the optimal solution of the empirical risk minimization problem* (2) *for a given set $S$ of $n$ training data points (i.i.d. from $\mathcal{D}$). Suppose there exists a distribution $\pi^\circ \in \mathcal{P}_2$ such that $h_{\pi^\circ}(x)y \geq 1/2$ for all $(x, y) \in \text{supp}\mathcal{D}$, and that $|h_\theta(x)| \leq 1$ holds for all $\theta \in \mathcal{H}$, $x \in \mathcal{X}$. Then, the following holds with probability $1 - \delta$ with respect to the choice of $S \subset \mathcal{X}$:*

$$\mathbb{E}_{(X,Y) \sim \mathcal{D}}[\ell_{01}(h_{\pi_*}(X), Y)] \leq \lambda_2 \text{KL}(\pi^\circ || \nu) + 8\sqrt{2}\sqrt{\frac{\text{KL}(\pi^\circ || \nu)}{n}} + 5\sqrt{\frac{1}{2n} \log \frac{1}{\delta}}.$$

The KL-term is bounded when $\pi^\circ$ is a variation of the Gaussian measure $\nu$ such that $\frac{d\pi^\circ}{d\nu} \log(\frac{d\pi^\circ}{d\nu})$ is integrable with respect to $\nu$. Note that if we set $\lambda_2 = 1/\sqrt{n}$, the generalization error is $O(1/\sqrt{n})$. This order matches that in the finite-dimensional input setting shown in Nitanda et al. (2021).

## 3 Mean-field optimization with infinite-dimensional Langevin algorithm

As previously remarked, the core idea of the mean-field analysis is to directly optimize the distribution of the parameters (2). For that purpose, the ideal situation would be to maintain infinitely many particles (neurons) to represent the distribution; however, in a tractable algorithm we need to use an approximation with finite number of particles. This finite-particle approximation may lead to instability due to the interaction between particles, which is known to be difficult to control.

To overcome this difficulty, we employ the *linearization technique* that was used in the original PDA and P-SDCA algorithms (Nitanda et al., 2021; Oko et al., 2022). We provide a short summary of this idea: given the $t$-th step solution $\pi_t$, we apply a first-order approximation of the loss function $L(\pi) := \mathbb{E}_{(X,Y) \sim \mathcal{D}}[\ell(\mathbb{E}_{\theta \sim \pi}[h_\theta(X)], Y)]$ as $(\pi - \pi_t) \frac{\delta L(\pi_t)}{\delta \pi} := \int \frac{\delta L(\pi_t)}{\delta \pi}(\theta) d(\pi - \pi_t)(\theta)$. Then, $\pi_{t+1}$ is updated as the minimizer of the linearized loss objective: $\arg\min_\pi (\pi - \pi_t) \frac{\delta L(\pi_t)}{\delta \pi} + \lambda_2 \text{KL}(\pi || \nu)$, which can be written as $d\pi_{t+1} = \exp(-\frac{1}{\lambda_2} \frac{\delta L(\pi_t)}{\delta \pi}) \cdot d\nu$. Hence we just need to draw particles from $\pi_{t+1}$ in the inner loop. Importantly, the interaction between particles disappear due to the linearization and we may sample in an i.i.d. manner. The original algorithms of PDA and P-SDCA for (finite-dimensional input) employed the gradient Langevin dynamics (GLD) and its variant such as MALA,

the convergence rate of which have been extensively studied (Raginsky et al., 2017; Vempala and Wibisono, 2019; Ma et al., 2019). However, in our setting we need to solve an infinite-dimensional sampling problem, which cannot be directly extended from the finite-dimensional counterpart.

Let $\pi$ be a distribution on $\mathcal{H}$ that satisfies $\frac{\mathrm{d}\pi}{\mathrm{d}\nu}(x) \propto \exp\left(-G(x)\right)$ (e.g., $G = \lambda_2^{-1}\delta L(\pi_t)/\delta\pi$). Then we know $\pi$ is the stationary distribution of the infinite-dimensional stochastic differential equation,

$$\mathrm{d}X_t = -\left(\nabla G(X_t) + AX_t\right)\mathrm{d}t + \sqrt{2}\mathrm{d}W_t,$$

where $(W_t)_{t\geq 0}$ is a cylindrical Brownian motion on $\mathcal{H}$ (Da Prato et al., 1996) and $\nabla G$ is the Riesz representer of the Fréchet derivative of $G$ on $\mathcal{H}$. To discretize the continuous-time SDE, we employ the semi-implicit Euler scheme:

$$X_{k+1} = X_k - \eta(\nabla G(X_k) + AX_{k+1}) + \sqrt{2\eta}\zeta_k,$$

i.e., $X_{k+1} = S_\eta\left(X_k - \eta\nabla G(X_k) + \sqrt{2\eta}\zeta_k\right)$, where $\zeta_k$ is a realization of standard Gaussian process on $\mathcal{H}$, and $S_\eta := (\mathrm{Id} + \eta A)^{-1}$ where $\mathrm{Id}$ is an identity mapping. We remark that our choice of semi-implicit scheme is due to the fact that $\zeta_k \notin \mathcal{H}$ in the infinite-dimensional setting, and thus the naive Euler-Maruyama discretization does not ensure $X_{k+1} \in \mathcal{H}$ while the semi-implict scheme does.

In practice, we need to approximate the infinite-dimensional vector $X_k$ by a finite-dimensional one. For that purpose, we apply the *Galerkin approximation*. Let $P_N$ be the orthogonal projection from $\mathcal{H}$ to $\mathcal{H}_N := \mathrm{span}\{e_0, \ldots, e_N\}$, and $\nabla G_N(x) := P_N(\nabla G(P_N x))$. Then the finite-dimensional approximation of the semi-implicit Euler scheme is given as

$$X_{k+1}^N = S_\eta\left(X_k^N - \eta\nabla G_N(X_k^N) + \sqrt{2\eta}P_N\zeta_k\right). \tag{3}$$

A convergence rate analysis of the infinite-dimensional gradient Langevin dynamics (3) with time discretization and finite-dimensional approximation was given by Muzellec et al. (2022), which is an important ingredient of our convergence rate analysis (the details can be found in Appendix B).

# 4  Infinite-dimensional Particle Dual Averaging

In this section, we present our first algorithm to solve the risk minimization problem (2) which is an infinite-dimensional extension of the Particle Dual Averaging (PDA) method Nitanda et al. (2021). PDA is a combination of Nesterov's dual averaging (Nesterov, 2009) and particle sampling; it is guaranteed to converge globally at a sublinear rate, but is tailored to the finite-dimensional setting. Extending the algorithm and convergence analysis to infinite-dimensional input is non-trivial, and to do so we make use of the weak convergence property of the infinite-dimensional gradient Langevin dynamics (GLD) shown in Muzellec et al. (2022).

**Algorithm description of infinite-dimensional PDA.** Here we provide the algorithm description of PDA with infinite-dimensional input. Since we are optimizing the expected loss directly, we employ an online gradient descent approach: we draw one input-output pair $(x_t, y_t)$ from $\mathcal{D}$ at the $t$-th iteration, and update the parameter by replacing the expected loss with a single-sample loss $\ell\left(\mathbb{E}_{\theta\sim\pi}[h_\theta(x_t)], y_t\right)$. In addition, we apply the following *linearization* to the loss at each outer loop step: suppose that we have $M$ particles $\Theta^{(t)} = (\theta_m^{(t)})_{m=1}^M$ whose empirical measure is denoted by $\hat{\pi}^{(t)} = \frac{1}{M}\sum_{m=1}^M \delta_{\theta_m^{(t)}}$ (i.e., $h_{\Theta^{(t)}} = \mathbb{E}_{\theta\sim\hat{\pi}^{(t)}}[h_\theta(\cdot)]$). Then the loss can be linearized around $\hat{\pi}^{(t)}$ as

$$(\pi - \hat{\pi}^{(t)})\frac{\delta\ell\left(\mathbb{E}_{\theta\sim\pi}[h_\theta(x_t)], y_t\right)}{\delta\pi}\bigg|_{\pi=\hat{\pi}^{(t)}} = \mathbb{E}_{\theta\sim\pi}\left[\partial_z\ell\left(\mathbb{E}_{\hat{\pi}^{(s)}}[h(\theta, x_s)], y_t\right)h(\theta, x_t)\right] + const.$$

We denote the gradient component in the right hand side as $g^{(t)}(\theta) := \partial_z\ell(h_{\Theta^{(t)}}(x^{(t)}), y_t)h(\theta, x_t)$. The idea of the Nesterov's dual averaging method is to take its weighted sum of the gradient over its history: $\bar{g}^{(t)}(\theta) := \frac{2}{\lambda_2(t+2)(t+1)}\sum_{s=1}^t sg^{(s)}(\theta)$, and update the distribution as

$$\pi_*^{(t+1)} := \underset{\pi}{\mathrm{argmin}}\ \ \mathbb{E}_{\theta\sim\pi}[\lambda_2\bar{g}^{(t)}(\theta)] + \lambda_2\mathrm{KL}(\pi||\nu).$$

As we have seen above, the Radon-Nykodym deriatives of $\pi_*^{(t+1)}$ with respect to $\nu$ is given by $\frac{\mathrm{d}\pi_*^{(t+1)}}{\mathrm{d}\nu}(\theta) \propto \exp\left(-\bar{g}^{(t)}(\theta)\right)$. Hence in the inner loop, we sample particles from $\pi_*^{(t+1)}$ using

infinite-dimensional gradient Langevin dynamics (3) by setting $G = \bar{g}^{(t)}(\theta)$. That is, we obtain the $M$-particle approximation by iteratively calculating the following updates:

$$\tilde{\theta}_{N,r}^{(k+1)} = S_{\eta_t}\left(\tilde{\theta}_{N,r}^{(k)} - \eta_t \nabla \bar{g}_N^{(t)}(\tilde{\theta}_{N,r}^{(k)}) + \sqrt{2\eta_t} P_N \zeta_k\right),$$

where $S_{\eta_t} := (\mathrm{Id} + \eta A)^{-1}$ and $\nabla \bar{g}_N^{(t)}(\cdot) := P_N(\nabla \bar{g}^{(t)}(P_N \cdot))$. We repeat the GLD iterations for $T_t$ times in the inner loop and obtain the set of particles $(\theta_m^{(t)})_{m=1}^M = (\tilde{\theta}_{N,m}^{(T_t)})_{m=1}^M$ for the next outer loop update. We denote by $\pi^{(t+1)}(\theta)$ the distribution of each particle $\tilde{\theta}_{N,m}^{(T_t)}$ (this is not the empirical distribution $\hat{\pi}_{t+1}$ but its "true" distribution). We will show that $\pi^{(t+1)}(\theta)$ well approximates the ideal update $\pi_*^{(t+1)}(\theta)$ in the proof of Theorem 5 below.

Finally, after $T$ outer loop iterations, the algorithm outputs $M$ parameters $\theta_1^{(\hat{t})}, \ldots, \theta_M^{(\hat{t})}$ where $\hat{t} \in \{2, \ldots, T+1\}$ is randomly chosen according to a distribution $\mathbb{P}(t) = \frac{2t}{T(T+3)}$ ($t = 2, \ldots, T+1$). The full algorithm is summarized in Appendix C.

**Convergence Analysis.** Now we establish the global convergence rate of our proposed infinite-dimensional PDA. Let $P_{\mathcal{X}}$ be the distribution of input $x \in \mathcal{X}$ and $\mathcal{G} := \mathrm{supp} P_{\mathcal{X}}$. Our analyses rely on the following assumptions.

**Assumption 2.**

*(A1)* $\mu_k \sim \frac{1}{k^2}$, *i.e. there exists a constant* $\alpha_1, \alpha_2$ *such that* $\alpha_1 \leq k^2 \mu_k \leq \alpha_2$.

*(A2)* $\mathcal{Y} \subset [-1, 1]$. *In addition,* $\ell(z, y)$ *is 1-smooth convex function with respect to* $z$. *Moreover, for all* $y, z \in \mathcal{Y}$, $|\partial_z \ell(z, y)| \leq 2$ *holds*.

*(A3) Both* $\sigma_1$ *and* $\sigma_2$ *are thrice continuously differentiable. Moreover, it holds that* $\max\{\|\sigma_1\|_\infty, \|\sigma_1'\|_\infty, \|\sigma_1''\|_\infty, \|\sigma_2\|_\infty, \|\sigma_2'\|_\infty, \|\sigma_2''\|_\infty\} \leq \hat{b}$.

*(A4) There exists* $\frac{1}{2} < \gamma < 2$, $B_{\mathcal{X}} > 0$ *such that* $\|x\|_{\mathcal{H}_{K^{1+\gamma}}} < B_{\mathcal{X}}$ *for all* $x \in \mathcal{G}$.

**Remark.** *(A1) is a sufficient condition to guarantee convergence of gradient Langevin dynamics (Muzellec et al., 2022), in which the exponent* $k^{-2}$ *may be generalized to* $k^{-p}$ ($p > 1$), *but we present the result only for* $p = 2$ *for simplicity. (A2) is satisfied by common loss functions such as square loss and logistic loss. (A3) is satisfied by several practical activation functions such as sigmoid and* $\tanh$. *Finally, (A4) is a regularity condition that is used in the convergence guarantee of GLD.*

We show the convergence of infinite-dimensional PDA by the following two steps: **(i)** bounding the difference between the optimal solution $\pi_{\mathrm{opt}}$ of the objective (2) and the optimal update $\pi_*^{(\hat{t})}$ at the $\hat{t}$-th iteration (Theorem 4), **(ii)** bounding the difference between the optimum auxiliary solution $\pi_*^{(\hat{t})}$ and its particle approximation $\hat{\pi}^{(\hat{t})}$ (Theorem 5).

To begin with, let $\epsilon_A^{(t)}$ be the weak convergence error of sampling in terms of the network output, i.e.,

$$\epsilon_A^{(t)} := \sup_{x \in \mathcal{G}} \left| \mathbb{E}_{\theta \sim \pi^{(t)}}[h(\theta, x)] - \mathbb{E}_{\theta \sim \pi_*^{(t)}}[h(\theta, x)] \right|.$$

This error can be bounded by the following theorem, proof of which can be found in Appendix C.1.

**Theorem 3.** *Under assumptions (A1)–(A4), for all* $t = 1, \ldots, T$ *and* $\kappa \in (0, 1/2)$, *there exists* $\hat{C}_1 = O\left((1 + \lambda_1^{-1})^2 \exp\left(O\left(\lambda_2^{-1}\right)\right)\right)$, $\hat{C}_2 = O\left(\exp\left(O\left(\lambda_2^{-1}\right)\right)\right)$, $\Lambda = \Omega\left(\min\{\lambda_1, \lambda_1^2 \lambda_2\} \exp\left(-O\left(\lambda_2^{-1}\right)\right)\right)$ *such that the following holds*

$$\epsilon_A^{(t)} \leq \hat{b}^2 \left(\hat{C}_1 \cdot \exp\left(-\Lambda\left(\frac{\eta_t}{\lambda_2} T_t - 1\right)\right) + \hat{C}_2\left(\mu_{N+1}^{1/2-\kappa} + \eta_t^{1/2-\kappa}\right)\right).$$

Theorem 3 establishes the convergence rate of sampling via infinite-dimensional GLD in the PDA algorithm. According to this theorem, to achieve sufficiently small sampling error $\epsilon_A^{(t)} < \epsilon$, it suffices to take $\mu_N = O(\epsilon^{2/(1-2\kappa)})$, $\eta_t = O(\epsilon^{2/(1-2\kappa)})$, $T_t = \Omega(\epsilon^{-\frac{2}{1-2\kappa}} \log(1/\epsilon)/\Lambda)$ with some $\kappa \in (0, 1/2)$. This result is comparable to the finite-dimensional counterpart in Nitanda et al. (2021),

where $\eta_t = O(\epsilon^2)$ and $T_t = \Theta(\epsilon^{-2} \log(1/\epsilon)/\Lambda)$ is shown to be sufficient, which is close to our bound up to a factor of $\kappa$ despite the infinite dimensionality. It is worth noting that the finite-dimensional analysis in Nitanda et al. (2021) is based on the convergence result of GLD under KL-divergence (Vempala and Wibisono, 2019). However, in our infinite-dimensional setting, we generally do not expect convergence under the KL metric since the distribution is hardly absolutely continuous with respect to another. Instead, our convergence guarantee is solely based on a weak convergence analysis, which allows us to establish computational complexity bound that is close to that of the finite-dimensional counterpart in Nitanda et al. (2021)[4].

As for **(i)**, the following theorem evaluates the difference between $\pi_*$ and $\pi_*^{(\hat{t})}$ with respect to the objective function value. The proof of this theorem can be found in Appendix C.2.

**Theorem 4.** *Suppose $\delta > 0$ and let $\bar{\epsilon}_A^{(T)} := \frac{2}{T(T+3)} \sum_{t=2}^{T+1} \epsilon_A^{(t)}$. Under **(A1)**, **(A2)**, **(A3)**, for all distributions $\pi_* \in \mathcal{P}_2$, the following inequality holds with probability $1 - \delta$:*

$$\frac{2}{T(T+3)} \sum_{t=2}^{T+1} t\Big(\mathcal{L}(\pi_*^{(t)}) - \mathcal{L}(\pi_*)\Big) \leq O\left(\bar{\epsilon}_A^{(T)} + \sqrt{\frac{\log\left(\frac{T}{\delta}\right)}{M}} + \frac{1 + \mathbb{E}_{\pi_*}[g^{(1)}]}{T^2} + \frac{\lambda_2 \cdot \mathrm{KL}(\pi_* \| \nu) + \frac{1}{\lambda_2}}{T}\right).$$

This theorem shows that when the number of outer iteration $T$ is larger than $O(1/\epsilon)$, the excess objective function value of the auxiliary solution $\pi_*^{(\hat{t})}$ becomes smaller than $\epsilon$. It is known that this $O(1/T)$ rate is the minimax optimal for online first-order methods when the objective is strongly convex (Agarwal et al., 2009). We note that the analysis in the original PDA algorithm (Nitanda et al., 2021) heavily relies on the existence of probability density function of the parameters; however, this is not guaranteed in the infinite-dimensional setting. We overcame this difficulty by reformulating the update using the Radon–Nikodym density with respect to the Gaussian measure $\nu$. Interestingly, this new technique yields a refined analysis that is indeed dimension independent.

Next we control the particle discretization error **(ii)**, i.e., the difference between the auxiliary solution and its particle approximation. The proof can be found in Appendix C.3.

**Theorem 5.** *Under **(A1)**, **(A2)**, **(A3)**, **(A4)**, the following inequality holds with probability $1 - \delta$:*

$$\sup_{x \in \mathcal{G}} \left| \mathbb{E}_{\theta \sim \pi_*^{(\hat{t})}}[h(\theta, x)] - \frac{1}{M} \sum_{r=1}^{M} h(\theta_r^{(\hat{t})}, x) \right| \leq O\left(\epsilon_A^{(\hat{t})} + \frac{B_{\mathcal{X}} \sqrt{\log B_{\mathcal{X}}} + \sqrt{\log\left(\frac{T}{\delta}\right)}}{\sqrt{M}}\right).$$

*In particular, we have $|L(\pi_*^{(\hat{t})}) - L(\hat{\pi}^{(\hat{t})})| = O\left(\epsilon_A^{(\hat{t})} + \frac{1}{\sqrt{M}}(B_{\mathcal{X}} \sqrt{\log B_{\mathcal{X}}} + \sqrt{\log\left(\frac{T}{\delta}\right)})\right)$.*

Notice that since the particle approximation does not have a density, the bound above does not control the objective $\mathcal{L}$ including the KL-regularization, but instead the $L^\infty$-norm error and the loss function $L$ of the estimated function. By combining the previous theorems, we know that when the number of outer loops iterations is $O(1/\epsilon)$, and the number of particles $M$ is $O((1/\epsilon)^2 \log(1/\epsilon))$, then our proposed algorithm achieves $\epsilon$-error in terms of the output of the model (or the loss function) from a solution with $\epsilon$-excess objective value.

Overall, the iteration complexity of our infinite-dimensional PDA to obtain $\epsilon$-accurate solution is $O(1/(\Lambda \epsilon^{\frac{3-2\kappa}{1-2\kappa}}) \log(1/\epsilon))$ for some $\kappa \in (0, 1/2)$, i.e., it is possible to obtain a solution with the desired accuracy at *polynomial order of iteration complexity* (but with exponential dependency on $1/\lambda_2$, which would be unavoidable for any mean-field method). We emphasize that our analysis guarantees the convergence to the global optimal solution, not the stationary point or local minimizer.

## 5 Infinite-dimensional Particle Stochastic Dual Coordinate Ascent

In the previous section, we considered the expected risk minimization problem. On the other hand, in the case of *empirical risk* minimization, we can improve the rate of convergence from polynomial to exponential by making use of the finite-sum property. For that purpose, we propose an infinite-dimensional extension of Particle Stochastic Dual Coordinate Ascent (P-SDCA) proposed in Oko et al. (2022). As in PDA, the original P-SDCA is also tailored to the finite-dimensional setting. Especially, the lack of density function in the infinite-dimensional parameter space necessitates a careful modification of the update rule. Indeed, we reconstruct the algorithm based on the density with respect to the Gaussian measure $\nu$.

---

[4]We also believe that the required $T_t$ in (Nitanda et al., 2021) can be improved to $\Theta(\epsilon^{-1} \log(1/\epsilon)/\Lambda)$.

**Algorithm description of infinite-dimensional P-SDCA.** Here we give an algorithmic description of our infinite-dimensional extension of P-SDCA. The main ingredient of SDCA is to execute optimization in the Fenchel dual of the objective function that is derived in the following lemma.

**Lemma 6.** *Suppose that $\ell(\cdot, y_i) : \mathbb{R} \to \mathbb{R}$ is a proper convex function and $h(\cdot, x_i) : \mathcal{H} \to \mathbb{R}$ is bounded for $i = 1, \ldots, n$. We denote the conjugate function of $\ell(\cdot, y_i)$ by $\ell^*(\cdot, y_i)$[5]. Define the dual objective $\mathcal{D} : \mathbb{R}^n \to \mathbb{R}$ as*

$$\mathcal{D}(g) := -\frac{1}{n} \sum_{i=1}^n \ell^*(g_i, y_i) - \lambda_2 \log \left( \mathbb{E}_{\theta \sim \nu} \left[ \exp \left( -\frac{1}{n\lambda_2} \sum_{i=1}^n g_i h(\theta, x_i) \right) \right] \right) \quad (g \in \mathbb{R}^n).$$

*Then, if $\inf_{\pi \in \mathcal{P}_2} \mathcal{L}(\pi) > -\infty$ holds, it holds that $\inf_{\pi \in \mathcal{P}_2} \mathcal{L}(\pi) = \sup_{g \in \mathbb{R}^n} \mathcal{D}(g)$.*

We construct the update rule based on this duality theorem. For a dual variable $g \in \mathbb{R}^n$, the corresponding primal solution can be retrieved by $p[g](\theta) := \frac{\exp\left(-\frac{1}{n\lambda_2} \sum_{i=1}^n g_i h(\theta, x_i)\right)}{\mathbb{E}_\nu \left[\exp\left(-\frac{1}{n\lambda_2} \sum_{i=1}^n g_i h(\theta, x_i)\right)\right]}$ (indeed, for the dual optimal solution $g^*$, the primal optimal solution can be recovered by $p[g^*]\mathrm{d}\nu$). In the SDCA algorithm, we randomly pick up one coordinate $i_t \in [n]$ from the uniform distribution, and update the corresponding coordinate so that it maximizes the dual objective. Suppose that we have the dual solution $g^{(t)}$ at the $t$-th iteration, then we update its $i_t$-th coordinate by the following formula where $i_t \in [n]$ is chosen uniformly at random:

$$g_{i_t}^{(t+1)} = \mathrm{argmax}_{g'_{i_t}} \left\{ -\ell^*(g'_{i_t}, y_{i_t}) + \mathbb{E}_\nu \left[ h(\theta, x_{i_t}) \cdot p[g^{(t)}](\theta) \right] \cdot \left( g'_{i_t} - g_{i_t}^{(t)} \right) - \frac{\left( g'_{i_t} - g_{i_t}^{(t)} \right)^2}{2n\lambda_2} \right\},$$

$$g_j^{(t+1)} = g_j^{(t)} \quad \text{for } j \neq i_t.$$

We can see that the update formula requires the expectation $\mathbb{E}_\nu \left[ h(\theta, x_{i_t}) \cdot p[g^{(t)}](\theta) \right]$. We approximate this expectation by a weighted average of $M$ particles as $\mathbb{E}_\nu \left[ h(\theta, x_{i_t}) \cdot p[g^{(t)}](\theta) \right] \approx \frac{\sum_{m=1}^M r_m^{(t)} h(\theta_m, x_{i_t})}{\sum_{m=1}^M r_m^{(t)}}$ where $r_m^{(t)}$ is the weight of the $m$-th particle. Here, the particles $(\theta_m)_{m=1}^M$ are "refreshed" once in every $\tilde{n}$ iterations, and in the remaining iterations, we only update the weightings of the particles $(r_m)_{m=1}^M$. In particular: (i) at the resampling steps (once in every $\tilde{n}$ iterations), we run the infinite-dimensional GLD to generate $M$ particles $(\theta_m)_{m=1}^M$ from the distribution $\pi_*^{(t)}$ that is given by $\frac{\mathrm{d}\pi_*^{(t)}}{\mathrm{d}\nu} = p[g^{(t)}]$, and reset the weights as $r_m = 1/M$ for all $m \in [M]$; (ii) in non-refreshing steps, we update the weights $(r_m)_{m=1}^M$ as $r_m^{(t+1)} = r_m^{(t)} \exp \left[ -\frac{h_{i_t}(\theta_m)(g_{i_t}^{(t+1)} - g_{i_t}^{(t)})}{n\lambda_2} \right]$. This re-weighting scheme allows us to reduce the number of sampling steps, which can be computationally demanding.

**Convergence analysis.** Now we establish the convergence rate of the proposed infinite-dimensional P-SDCA. Suppose at the $T$-th sampling stage, we run the GLD with step size $\eta_T$ for $J_T$ iterations. Then the sampling error can be bounded as follows: denote by $p^{(\tilde{n}T)}$ the distribution of the particles, under **(A1),(A2),(A3),(A4)**, for all $\kappa \in (0, 1/2)$, it holds that

$$\left| \mathbb{E}_{\theta \sim p^{(\tilde{n}T)}} [\phi(\theta)] - \mathbb{E}_\nu \left[ \phi(\theta) p[g^{(\tilde{n}T)}](\theta) \right] \right| \leq \check{C}_1 \mathrm{e}^{-\Lambda \left( \frac{\eta_T}{\lambda_2} J_T - 1 \right)} + \check{C}_2 \left( \mu_{N+1}^{1/2 - \kappa} + \eta_T^{1/2 - \kappa} \right) =: \epsilon_C^{(\tilde{n}T)},$$

where $\check{C}_1, \check{C}_2, \Lambda$ are constants depending on $\hat{b}, \lambda_1, \lambda_2$ and $\phi : \mathcal{H} \to \mathbb{R}$ is any test function satisfying $\|\phi\|_\infty \leq 1$ with sufficient smoothness (see Appendix D.3 for more details). Using the sampling error $\epsilon_C^{(\tilde{n}T)}$, we can now derive the iteration complexity of the algorithm.

**Theorem 7.** *Assume (A1)–(A4) and $|\ell(x, y_i) - \inf_{x'} \ell(x', y_i)| \leq B_1$ for all $x \in [-\hat{b}, \hat{b}]$ and $i \in [n]$. Let $\tilde{s} := \frac{\tilde{n}\lambda_2}{1 + n\lambda_2}$ and take any $T_{\mathrm{end}} \geq 2\frac{n}{\tilde{n}} \left( 1 + \frac{1}{n\lambda_2} \right) \log \left( \left( n + \frac{1}{\lambda_2} \right) \frac{B_1 + 2 + \frac{1}{1 - \exp(-\tilde{s})}}{\epsilon_P} \right)$ for $\epsilon_P > 0$. Suppose that $\epsilon_C^{(\tilde{n}T)}$ and $M$ satisfy $\epsilon_C^{(\tilde{n}T)} \leq \hat{C}_1 \exp \left( -\frac{\tilde{s}T}{2} \right) (\forall T \in [T_{\mathrm{end}}])$ and $M \geq \frac{\hat{C}_2^2}{(\epsilon_C^{(\tilde{n}T)})^2} \log \left( \frac{4n T_{\mathrm{end}}}{\delta} \right)$ where $\hat{C}_1^{-1} = O \left( \lambda_2^{-\frac{1}{2}} \frac{\hat{C}_2}{1 + \frac{1}{n\lambda_2}} \exp \left( \frac{\tilde{n}(2\hat{b}\hat{C}_2 + 1)}{n\lambda_2 + 1} \right) \right)$ and $\hat{C}_2 = \exp \left( \frac{8 \max\{\hat{b}^2, \hat{b}^4\}\tilde{n}}{\lambda_2 n} \right)$. Then, after $\tilde{n}T_{\mathrm{end}}$ iterations of infinite-dimensional P-SDCA, the duality gap can be bounded as $\mathbb{E} \left[ \mathcal{L}(p[g^{(\tilde{n}T_{\mathrm{end}})}]) - \mathcal{D}(g^{(\tilde{n}T_{\mathrm{end}})}) \mid \mathcal{E} \right] \leq \epsilon_P$, for a high-probability event $\mathcal{E}$ such that $P(\mathcal{E}) \geq 1 - \delta$.*

---

[5]The convex conjugate of a convex function $f : \mathbb{R} \to \mathbb{R}$ is given by $f^*(u) = \sup_{x \in \mathbb{R}} \{xu - f(x)\}$.

The detailed statement and proof of Theorem 7 can be found in Appendix D.4. Here, the expectation is taken over the choice of the coordinates in each step and the event $\mathcal{E}$ corresponds to measure concentration induced by the $M$-particle sampling. Consequently, P-SDCA can achieve a duality gap smaller than $\epsilon_P$ with $O(\log(1/\epsilon_P))$ outer loop iterations. In combination with the sampling complexity, we see that the total iteration cost to obtain a solution with $\epsilon_P$ duality gap is $O((1/\epsilon_P)^{\frac{2}{1-2\kappa}}\log(1/\epsilon_P))$, which is much better than that of PDA shown in Section 4.

## 6  Numerical Experiments

We numerical evaluate our proposed methods on a non-linear functional regression problem. Let $\mathcal{H}$ be $L^2([0,1])$, which is a set of square integrable functions defined on $[0,1]$, and $\mathcal{H}_K$ be the Sobolev space $H^1([0,1])$, which is a set of absolutely continuous functions defined on $[0,1]$ satisfying $u(0) = 0$ and $u' \in L^2([0,1])$. Then, $\mathcal{H}$ is a Hilbert space equipped with inner product defined by $\langle x_1, x_2 \rangle_{\mathcal{H}} = \int_0^1 x_1(t)x_2(t)\mathrm{d}t$, and $\mathcal{H}_K$ is a Hilbert space equipped with inner product defined by $\langle x_1, x_2 \rangle_{\mathcal{H}_K} = \int_0^1 x_1'(t)x_2'(t)\mathrm{d}t$. Consequently, $\mathcal{H}_K$ is a reproducing kernel Hilbert space whose kernel function is given as $K(s,t) = \min\{s,t\}$.

We consider a teacher-student setup. The data is generated as $y_i = \hat{\sigma}_1(a^\circ)\hat{\sigma}_2(\langle w^\circ, x_i \rangle_{\mathcal{H}}) + \epsilon_i$ $(i = 1, \ldots, n)$ where $\epsilon_i \sim N(0,1)$ and $\hat{\sigma}_1(\cdot) = \hat{\sigma}_2(\cdot) = \mathrm{sign}(\cdot)$. The true parameters $(a^\circ, w^\circ) \in \mathcal{H}$ are generated by $a^\circ \sim N(0,5^2)$ and $w^\circ = \sum_{j=1}^{\bar{N}} w^{(j)}e_j$, where $w^{(j)} \sim N(0,5^2)$. We randomly generate $x_i$ from a Gaussian process whose covariance operator is a kernel function of $\mathcal{H}_{K^3}$, and accept it when it holds that $\|x_i\|_{\mathcal{H}_{K^2}} < B_{\mathcal{X}}$ with some $B_{\mathcal{X}}$. In our implementation, we only used $\bar{N}$ bases of $\mathcal{H}$ to define $w^\circ$ (note that $w^\circ \in \mathcal{H}_0$ still holds for this implementation). We set $\bar{N}$ so that it is larger than $N$ that is the number of bases for the Galerkin approximation.

In addition to demonstrating the validity of our proposed method, we also illustrate the benefit of feature learning. For that purpose, we compare the performance of the two-layer mean-field neural network against that of non-adaptive estimators: the ridge regression estimator and the Nadaraya-Watson estimator. Here, we use the terminology "non-adaptive" to indicate that they do not perform feature learning. For ridge regression, we implemented the estimator using an $N$-basis approximation, i.e., $y = \langle \sum_{j=0}^N \alpha_j e_j, x \rangle_{\mathcal{H}}$, where $(\alpha_1, \ldots, \alpha_N) \in \mathbb{R}^N$ are chosen to minimize $\frac{1}{n}\sum_{i=1}^n \left( \sum_{j=1}^N x_i^{(j)}\alpha_j - y_i \right)^2 + \lambda \sum_{j=1}^N \alpha_j^2$ with regularization parameter $\lambda > 0$. As for the Nadaraya-Watson estimator, the estimator $\hat{y}$ for an input $x$ is given by $\hat{y} = \frac{\sum_{i=1}^n y_i k(\|x_i - x\|_{\mathcal{H}}/h)}{\sum_{i=1}^n k(\|x_i - x\|_{\mathcal{H}}/h)}$, where $h > 0$ is a bandwidth and $k(u) = \max\{1 - u^2, 0\}$ for $u \in \mathbb{R}$ (Epanechnikov kernel).

We used the squared loss for the optimization of two-layer neural network. We employed $\tanh$ for the activation functions $\sigma_1, \sigma_2$ of the student model. Hence, the teacher model is a bit out side of the student model. Our experiments were performed using the following hyperparameters: Upper bound of input data in $\mathcal{H}_{K^2}$: $B_{\mathcal{X}} = 100$; Number of basis functions to generate data: $\bar{N} = 300$; Number of training data $n = 100$; Number of basis functions for Galerkin approximation: $N = 150$; Regularization parameters for PDA and P-SDCA: $\lambda_1 = 10^{-2}$, $\lambda_2 = 10^{-5}$; Number of particles: $M = 200$; Hyperparameters for PDA: $T_t = 10$, $\eta_t = 10^{-5}$, minibatch size 50;

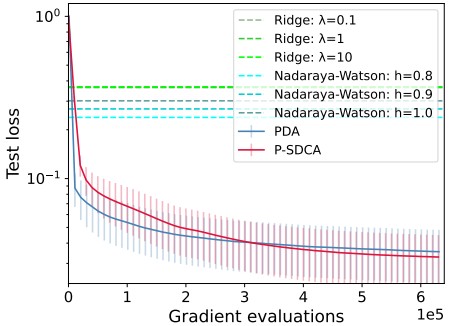

Figure 1: gradient calculations vs. test loss.

Hyperparameters for P-SDCA: $J_T = 10$, $\tilde{n} = 1000$, $\eta_t = 10^{-4}$; Regularization parameter for ridge regression: $\lambda = 0.1, 1, 10$ (we observed that $\lambda = 1$ achieved the best performance over all choices of $\lambda$); Bandwidth for Nadaraya-Watson estimator: $h = 0.8, 0.9, 1.0$.

As shown in Figure 1, PDA and P-SDCA converge and achieve small test loss. Moreover, for both algorithms, the eventual test loss is smaller than those of linear estimators (Ridge-regression and Nadaraya-Watson estimator), which illustrates that the mean-field two-layer neural network has better generalization ability than non-adaptive methods. This is mainly due to the feature learning ability of the mean-field networks, whereas the non-adaptive methods are not able to find informative features.

# 7 Conclusion

In this work we studied two-layer neural networks in the mean-field regime as a model for machine learning problems with infinite-dimensional input. We proposed two optimization algorithms to learn the neural network, by extending the Particle Dual Averaging and Particle Stochastic Dual Coordinate Ascent method to infinite-dimensional parameter space. Leveraging the convergence guarantee of infinite-dimensional gradient Langevin dynamics, we showed that our proposed methods can globally optimize the training objective at a rate of polynomial order for the online setting and exponential order for the finite-sum setting. Numerical experiments on synthetic data confirmed that mean-field neural network outperforms linear estimators that do not learn features.

## Acknowledgments

NN was partially supported by CREST (JPMJCR2015). TS was partially supported by Japan Digital Design and CREST (JPMJCR2115). AN was partially supported by JSPS Kakenhi (22H03650) and JST-PRESTO (JPMJPR1928). DW was partially supported by the Borealis AI Fellowship.

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
