# OpenReview forum: "Two-layer neural network on infinite dimensional data:  global optimization guarantee in the mean-field regime"
_NeurIPS.cc/2022/Conference — NeurIPS 2022 Accept_

### Official Review · Reviewer_JAao · 2022-07-10

**Rating:** 9
**Confidence:** 3
**Soundness:** 4 excellent
**Presentation:** 4 excellent
**Contribution:** 4 excellent

**Summary:**

This paper studies the  mean field limit of two-layer neural network in an infinite-dimensional parameter space. The main contributions are:

- A generalization error bound of the regularized empirical risk minimizer.

- Two optimization algorithms for our infinite-dimensional neural network by incorporating the  infinite-dimensional Langevin dynamics in the inner loop of PDA and P-SDCA.

- Theoretical guarantees for the convergence of the two algorithms.

**Questions:**

I only have minor comments and questions.


- Are the two distributions $P_{\mathcal{X}}$ and $P_{X}$ in line 222 the same?

- Can the author comment on the choice of the Gaussian measure as the reference measure in the regularization? In the finite dimensional case, obviously the flat Lebesgue measure is used as the reference. In the infinite dimensional case, I understand that Gaussians are the canonical choice. However, does the choice of Gaussian measure affect the generalization error? I would image the covariance of the Gaussian reference is essential since it determines the smoothness of functions that the Gaussian measure is supported.

- Is the Nesterov’s dual averaging essential for the main results (Theorems 3-5) to hold? I am not an expert on stochastic optimization, but is it possible to establish these results if one only use the  gradient information at the most recent step?


**Ethics Review Area:**

["I don’t know"]

**Limitations:**

I am not aware of  potential negative societal impact of this work. The author did not discuss the limitations explicitly in the conclusion, but comment on some limitations of the main results after the statements.

**Strengths And Weaknesses:**

- Strengths. This is a very nice (and perhaps the first) paper on the training analysis of  two-layer neural network on an infinite-dimensional space in the mean field regime. The extension from finite dimensional setting to infinite dimensions is highly nontrivial.  It is among the best of all papers I have reviewed in NeurIPS 2022.

 - Weaknesses. In my opinion, the paper is strong enough to be accepted. I do not have comments on the weakness, except minor comments which requires some clarifications; see below.

---

> ### Author Response · Authors · 2022-08-02
> **Answer to Reviewer JAao**
>
> Thank you for your positive feedback and valuable comments. We address the technical comments below.
>
> **1.  ``Are the two distributions $P_\mathcal{X}$ and $P_X$ in line 222 the same?”**
>
> Thank you for your pointing out this typo. We have updated the manuscript accordingly.
>
> **2. ``Can the author comment on the choice of the Gaussian measure as the reference measure in the regularization? ...  However, does the choice of Gaussian measure affect the generalization error?”**
>
> Indeed in an infinite dimensional setting, we need to impose some kind of model inductive bias to obtain an informative estimator.
> In our setting the choice of kernel (the covariance operator) is the most important part, as the spectral property $\mu_k$ controls the ``capacity'' of the model. In our paper, we considered $\mu_k \sim k^{-2}$ for simplicity, but this choice can be relaxed to more general settings such as $\mu_k \sim k^{-p}$, in which case the decay rate $p$ should affect the generalization performance. This is known as the *source and capacity condition* in the literature of kernel ridge regression, and its effect on generalization performance has been extensively studied. We expect a similar impact in the infinite-dimensional mean field setting too.
>
> In addition, the generalization gap in our Theorem 2 is controlled by the KL-divergence between $\nu$ and a target distribution $\pi^\circ$. In that sense, how well the reference distribution $\nu$ is ``aligned'' with the target distribution also affects the generalization, i.e., better alignment yields better generalization.
>
> **3. ``Is the Nesterov’s dual averaging essential for the main results (Theorems 3-5) to hold?”**
>
> The main reason that we chose the Nesterov's dual averaging is because it is easier to guarantee the boundedness of the intermediate solution than the vanilla SGD. This is parallel to the finite-dimensional setting, where it is known that the dual averaging method requires weaker condition to guarantee the boundedness on the intermediate solution.
> We note that the boundedness of intermediate solutions leads to reasonable sampling complexity in our problem settings.
> We chose SDCA (rather than SAGA and SVRG) as an exponentially converging method in a finite sum setting for the same reason.
> It is an interesting future work to clarify whether the last gradient type method would also converge in our problem setting.
>
> We would be happy to clarify any concerns or answer any questions that may come up during the discussion period.

---

### Official Review · Reviewer_EsYc · 2022-07-11

**Rating:** 5
**Confidence:** 3
**Soundness:** 3 good
**Presentation:** 3 good
**Contribution:** 2 fair

**Summary:**

In this paper, the authors extend the mean-field limit in the two-layer neural networks with finite-dimensional input to the infinite-dimensional input setting. The generalization bound was derived to show the regularized empirical risk minimizer is provable generalized at rate of $O(1/\sqrt{n})$. Two optimization algorithms (PDA, P-SDCA) based on gradient Langevin dynamics are proposed. Authors proved that two algorithms could converge to optimal solution under certain assumptions. Experiment results also support the theoretical analysis.

**Questions:**

1. In Theorem 7, at the last line what is the expectation taken over? Is it taken over the random noise during training?
2. It would be better if authors could write a main theorem for PDA to combine Theorem 3,4,5 (just like Theorem 7 for P-SDCA).


**Limitations:**

Authors discuss the limitations of current work. As far as I could see, there are no potential negative social impacts.

**Strengths And Weaknesses:**

Strengths
1. In general the paper is well organized and clearly written.
2. Extending mean-field limit to the infinite-dimensional input setting to allow feature learning seems to be new. The two proposed optimization algorithms as well as the generalization bound also seem to be new in the literature.
3. The theoretical results are well supported with proof ideas in the main text and complete proof in the appendix.
4. This paper gives generalization bound $O(1/\sqrt{n})$ that shows regularized empirical risk minimizer generalizes provided enough samples in the infinite-dimensional data setting.

Weaknesses
1. While the two proposed optimization algorithms could converge to optimal solution, their running time are exponential in regularization parameter $\lambda_2$. As suggested by Thm 1, $\lambda_2$ should be set to $1/\sqrt{n}$, which implies an exponential runtime. This seems unavoidable in the general mean-field setting with gradient Langevin dynamics. However, it would be better if authors could clearly state this, since the current abstract and contribution part seems to suggest the runtime is polynomial.
2. The generalization bound depends on $KL(\pi^o||\nu)$ where $\pi^o$ is the target distribution and $\nu$ is Gaussian distribution. It would be better if some discussion could be added to clarify in some common settings when this KL divergence is bounded (e.g., $O(1)$) and when it is not.

---

> ### Author Response · Authors · 2022-08-02
> **Answer to Reviewer EsYc**
>
> Thank you for your thoughtful comments. We address the technical comments below.
>
> **1. ``Running time are exponential in regularization parameter $\lambda_2$”**
>
> Indeed as you pointed out, the convergence rate has an exponential dependency on $1/\lambda_2$; this limitation is emphasized on line 282 (line 286 of the revised version), and for similar reasons, in the abstract and introduction we stated that our result holds for the regularized objective.
> Following your suggestion, we have added one sentence to highlight this point in the introduction of the revised version.
>
> **2. ``The generalization bound depends on $\mathrm{KL}$."**
>
> Thank you for the suggestion.
> The necessary and sufficient condition for the bounded $\mathrm{KL}$-term is that the target distribution is a bounded variation of the reference Gaussian measure, that is, $\mathrm{d}\pi^\circ \sim h(\cdot)\mathrm{d}\nu$ with a function $h$ such that $h\log(h)$ is integrable with respect to $\nu$. Especially, if $h$ is bounded from below and above, then such boundedness holds.
> We have added this explanation in the revised version.
>
> In addition, although we stated the assertion under a simple assumption that $\pi^\circ$ yields perfect separation, we may relax this condition in exchange for some additional terms representing the error of $\pi^\circ$. However, we decided to present the current simple form for conciseness and readability.
>
> **3. ``In Theorem 7, at the last line what is the expectation taken over? Is it taken over the random noise during training?”**
>
> The expectation is taken over the choice of coordinates randomly selected in each step.
> The randomness induced by sampling the $M$ particles is condensed by the definition of the event $\mathcal{E}$. The definition of $\mathcal{E}$ is given in line 939 of the supplementary material (line 944 of the revised version).
> We omitted the details in the main text due to the space limitation, but now we have added a one-sentence explanation in the revised version.
>
> **4. ``It would be better if authors could write a main theorem for PDA to combine Theorem 3,4,5 (just like Theorem 7 for P-SDCA).”**
>
> Thank you for your constructive comment.
> As you said, it would be desirable to state a theorem by combining Theorems 3,4, and 5, and such a combination is possible by considering the convergence of distributions of finite-particles. However, such a unified statement would be rather opaque and mathematically involved. Moreover, the KL-regularization-term is not well defined for the discrete distribution $\hat\pi_t$ that is obtained by our particle approximation method. Therefore, for simplicity we decided to show Theorems 4 and 5 separately where Theorem 4 asserts convergence of the whole objective with respect to the intermediate target $\pi_*^{(t)}$, and Theorem 5 asserts that the solution from our algorithm well approximates the neural network corresponding to the intermediate target.
> Since the evaluation of discretization error in Theorem 3 is required both for Theorems 4 and 5, we think Theorem 3 has its own importance and thus it is presented as a separated statement. Also, to compensate for the fragmented theorems, we presented the total amount of computation required to reach $\epsilon$-accurate solution in line 280 (line 284 of the revised version).
>
> We would be happy to clarify any concerns or answer any questions that may come up during the discussion period.

---

### Official Review · Reviewer_dM1S · 2022-07-27

**Rating:** 6
**Confidence:** 2
**Soundness:** 3 good
**Presentation:** 3 good
**Contribution:** 3 good

**Summary:**

This submission studies the optimization of the neural network with infinite-dimensional input in the mean-field regime and provides a convergence guarantee based on weak convergence. This result utilizes the proximal gradient method which solves the optimization task by linearization and Langevin sampling approximation. The authors study the convergence rate in both finite-sum and online settings. Compared to previous work, this submission considers the infinite-dimensional case and makes use of the analysis in infinite-dimensional Langevin. Besides, the authors provide the experiment on functional regression. The primary contribution is the convergence guarantee of the algorithm with infinite-dimensional input.

**Questions:**

Refer to strengths and weaknesses.

**Limitations:**

Refer to strengths and weaknesses

**Strengths And Weaknesses:**

Strengths:

This paper provides a complete and detailed analysis on infinite-dimensional-input P-SDCA and PDA and provides the weak convergence result. First the paper is well-written and the analysis is rigorous and well-orginized. The assumptions are stated explicitly. The authors also llustrate the effectiveness by numerical experiments.

Weakness:

Lack disscussion on the theoretical result. It would be better to discuss the implication of the convergence rate e.g. on $\lambda_2$. I am not familiar with this field but it seems that this line of research cannot overcome the exponential dependency on sample size M which makes the bound less informative.

Minor:

In proof of thm 4, drop the weight t.

---

> ### Author Response · Authors · 2022-08-02
> **Answer to Reviewer dM1S**
>
> Thank you for your helpful comments. We address the technical comments below.
>
> **1. ``Lack disscussion on the theoretical result. It would be better to discuss the implication of the convergence rate e.g. on $\lambda_2$.''**
>
> As commented on line 282 (line 286 of the revised version), the exponential dependence of the iteration complexity on $1/\lambda_2$ is generally unavoidable in our mean-field setting (this is consistent with the original algorithm PDA and P-SDCA for for finite-dimensional input settings).  To elaborate on this point, our statement that exponential dependence on $1/\lambda_2$ is unavoidable is based on the fact that there are examples where the Langevin dynamics require exponentially many steps to get out of a local optimal solution as shown in [Berglund, 2011]. In those examples, the iteration complexity of Langevin algorithms depends exponentially on the noise variance relative to the variation of the target function (please refer to [Menz and Schlichting, 2012]).
> The original PDA and P-SDCA papers utilized the convergence of Langevin algorithm under the KL divergence shown in [Vempala and Wibsono, 2019], which crucially relies on the log-Sobolev inequality which has a constant exponentially dependent on $\lambda_2$.
> In our study, we utilized a different weak convergence result due to [Muzellec et al., 2022], which also has exponential dependence on the inverse temperature parameter (essentially the same as $1/\lambda_2$), and hence we cannot avoid this shortcoming as well.
>
> **2.  ``In proof of thm 4, drop the weight t.”**
>
> Thank you for pointing this out.
> Since we are taking the weighted average over $t=2,\dots,T+1$ in the left hand side,
> the right hand side should also be a weighted average so that $\epsilon_A^{(t)}$ should be replaced by $\frac{2}{T(T+3)}\sum_{t=2}^{T+1}\epsilon_A^{(t)}$.
> We have fixed this point in the revised version.
>
> We would be happy to clarify any concerns or answer any questions that may come up during the discussion period.
>
> ---
>
> [Berglund, 2011] Kramers' law: Validity, derivations and generalisations.
>
> [Menz and Schlichting, 2012] Poincaré and logarithmic Sobolev inequalities by decomposition of the energy landscape.

---

### Meta-Review · Area_Chair_bHnK · 2022-08-23

**Recommendation:** Accept
**Confidence:** Certain

**Metareview:**

The paper considers the two-layer neural network in the mean-field regime and proposes an algorithm with complexity independent of the input dimension. Overall, I think the paper is very interesting. I recommend an acceptance.

**Award:**

No

---

### Decision · Program_Chairs · 2022-09-14

Accept